# First Record of the Family Malachiidae (Coleoptera: Cleroidea) from Mid-Cretaceous Burmese Amber with a Description of *Burmalachius acroantennatus* Gen. et Spec. Nov.

**DOI:** 10.3390/life13091938

**Published:** 2023-09-20

**Authors:** Sergei E. Tshernyshev, Andrei A. Legalov

**Affiliations:** 1Institute of Systematics and Ecology of Animals, Siberian Branch, Russian Academy of Sciences, Frunze Str., 11, 630091 Novosibirsk, Russia; fossilweevils@gmail.com; 2Biological Institute, Tomsk State University, Lenina Prospekt, 36, 634050 Tomsk, Russia; 3Department of Ecology, Biochemistry and Biotechnology, Altai State University, 656049 Barnaul, Russia

**Keywords:** Cleroidea, Malachiidae, Malachiini, fossil soft-winged flower beetle, Myanmar, Cenomanian

## Abstract

A new soft-winged flower beetle, *Burmalachius acroantennatus* gen. et sp. nov. belonging to the tribe Malachiini (Coleoptera: Malachiidae), discovered in mid-Cretaceous Burmese amber is described. The new genus differs from the congeners of the tribe Malachiini in possessing the following characteristics: anterior tibiae widened and slightly curved inwards and excavate near the apices, tarsomeres of anterior legs depressed, 1st to 3rd tarsomeres simple and almost of equal size, tarsal comb lacking, “clavate” antennae due to dilated three apical antennomeres, 1st and 2nd antennomeres enlarged and of identical size, head strongly elongate, sides of elytra with carinate margins and widened epipleurae. This is the first record of the family Malachiidae in Burmese amber. Illustrations of the newly described species are provided. Keys for the identification of Melyrid lineage families, subfamilies of Malachiidae, tribes of the subfamily Malachiinae and genera of the tribe Malachiini are also given.

## 1. Introduction

Beetles of the superfamily Cleroidea that include the melyrid lineage of families Melyridae, Dasytidae and Rhadalidae are well known from the Eocene Epoch, the second of three major worldwide divisions of the Paleogene. Beetles have already been described in Baltic and Rovno amber and from imprints in Florissant deposit [1,2,3,4,5,6,7,8,9,10,11,12,13,14,15,16,17,18,19]. Detailed reviews of the composition of the amber faunas have been provided for the families Dasytidae [14,16] and Malachiidae [15,17]. In fact, the earliest records of Malachiidae beetles are from the Eocene Epoch (66 to 23 million years ago). In contrast, the Burmese amber inclusion mined in Myanmar is c. 100 million years old and belongs to the latest Albian to earliest Cenomanian ages of the mid-Cretaceous period [20]. Only one “Melyrid” beetle, *Cretaidgia burmensis* Zhao, Liu & Yu, 2021 [21] of the family Prionoceridae, has been described in Burmese amber. A new inclusion in Burmese amber attracts attention since its body is widened posteriorly and flattened, as is typical of Malachiidae. Although the dilated apical antennomeres and elongate head are similar to representatives of Prionoceridae, the modified anterior legs and small, but visible, thoracic vesicles are typical of the family Malachiidae. However, some characteristics typical of Prionoceridae also occur in some Malachiidae species, such as dilated apical antennomeres in *Acromalachius* Evers, 1985, and elongate head in *Tanaops* LeConte, 1859, but characteristics typical of Malachiidae, such as thoracic vesicles or special dilated and excavate anterior tibiae in males, have never been found in Prionoceridae.

Malachiid imagoes differ in their morphology, the males of most species being provided with specific structures, such as protrusions and depressions, lamellar and spicular appendages, dilated and modified extremities or antennomeres, and tufts of hairs and setae on different parts of the body. The male from the Burmese amber possesses special characteristics, the anterior tibiae being distally dilated and excavated, and the basal tarsomeres of the anterior tarsi being wide and depressed.

All species of Malachiidae described from fossil remains of males belong to three tribes, namely Troglopini (*Protocephaloncus* Tshernyshev, 2016), Palpattalini (*Palpattalus* Tshernyshev, 2016, *Palpattalusinus* Tshernyshev, 2020, *Aliattalus* Tshernyshev, 2021) and Malachiini (*Premalachius* Tshernyshev, 2020) [15,17,22]. Most of the species listed in old papers or recently described on the basis of a single female need additional study to define their tribal attribution (see Tshernyshev [22]). The new genus and species, *Burmalachius acroantennatus* gen. et spec. nov., described below, is attributed to the tribe Malachiini on the basis of the size and shape of the tarsomeres and antennomeres.

## 2. Materials and Methods

The specimen from Burmese (Kachin) amber originated from the Noije Bum Summit Site mine in the Hukawng Valley (southwest of Maingkhwan, Kachin State (26°20′ N, 96°36′ E), northern Myanmar). A Zircon U-Pb and trace element analyses of amber from different locations in Myanmar confirmed an age of around 100 Ma for amber from the Hukawng Valley, as well as an age range of 72 to 110 Ma for amber from other sites in northern Myanmar [20]. Based on paleontological evidence, the site was dated to the late Albian age of the Early Cretaceous period [23], placing its age at 97–110 million years ago.

For descriptions, special male structures and genitalia were studied. The term “special male structures” is not analogue to the term “Excitatoren”, that means a different kind of structures located in different parts of the male body of soft-winged flower beetles and bearing ducts of pheromone glands necessary for female attraction and successful copulation [24,25,26,27]. The “special male structures” includes all typical parts of males, irrespective of providing them with pheromone glands or not. In the new species, widened and emarginate anterior tibiae, dilated basal tarsomeres in anterior legs, and enlarged apical antennomeres are considered as special male characteristics. The terminology of terminalia morphology is according to Lawrence et al. [28], namely (in comparison with previously used terms), pygidium for apical tergite, and ultimate abdominal ventrite for apical sternite.

The holotype of the new species is housed in S.E. Tshernyshev’s collection at the Institute of Systematics and Ecology of Animals SB RAS, Novosibirsk, Russia (ISEZh).

The beetle was studied using an Amscope trinocular stereomicroscope (Ultimate Trinocular Zoom Microscope 6.7X-90X Model ZM-2TY) and digital photographs were taken using a Carl Zeiss Stemi 2000 trinocular microscope and the AxioVision programme.

Nomenclatural acts introduced in the present work are registered in ZooBank (www.zoobank.org accessed on 18 September 2023) under LSID urn:lsid:zoobank.org:pub: LSID urn:lsid:zoobank.org:pub:BB6ADB10-6F31-4F0B-832C-90C69215BE2C.

## 3. Results


**Systematic**


Family **Malachiidae** Fleming, 1821

Subfamily **Malachiinae** Fleming, 1821

Tribe **Malachiini** Fleming, 1821

Genus ***Burmalachius*** gen. nov. (Figure 1 and Figure 2)

urn:lsid:zoobank.org:act:19D0ED41-447F-4BE0-8AD0-E738A4FBC651

Type species. *Burmalachius acroantennatus* sp. nov.

**Figure 1 life-13-01938-f001:**
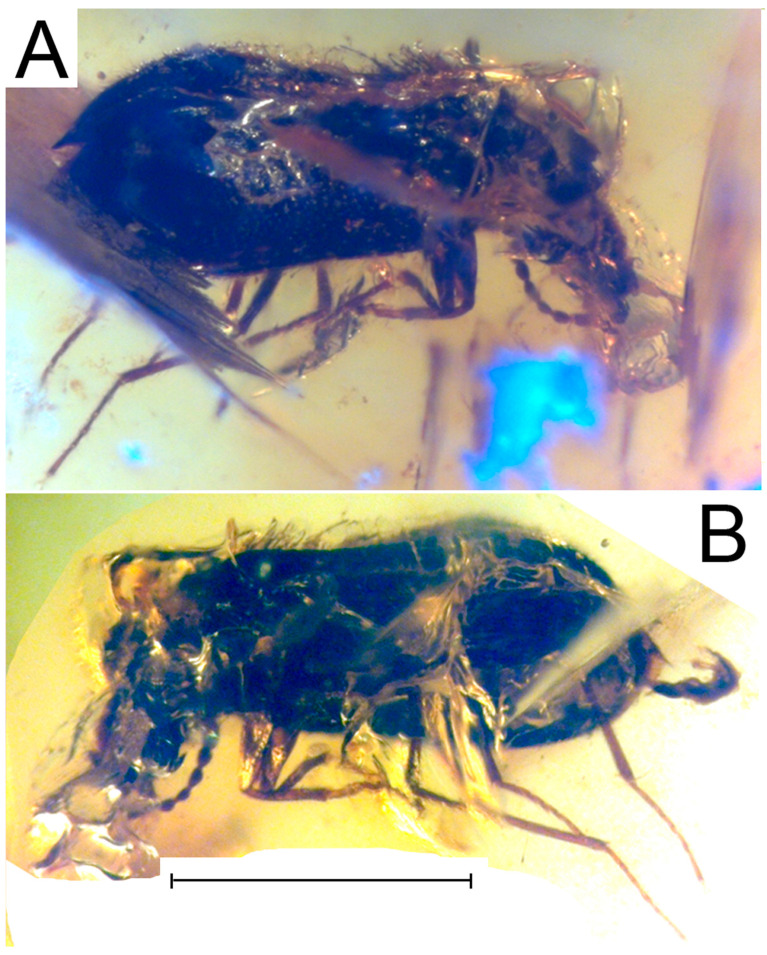
*Burmalachius acroantennatus* gen. et sp. nov., photo of holotype, male. (**A**) External appearance, dorsal view. (**B**) External appearance, ventral view. Scale bar 1 mm.

**Figure 2 life-13-01938-f002:**
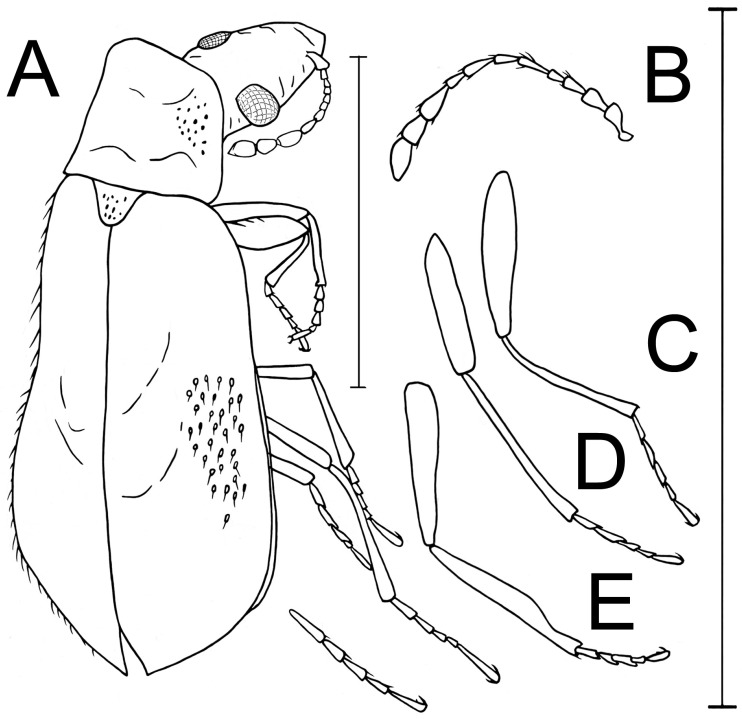
*Burmalachius acroantennatus* gen. et sp. nov., figures of holotype, male (**A**–**E**). (**A**) External appearance. (**B**) Right antenna. (**C**) Anterior leg. (**D**) Intermediate leg. (**E**) Posterior leg. Scale bars 1 mm.

*Etymology*. The name draws attention to the similarity of the new genus with the genera of the tribe Malachiini and is composed of two words, Burma, the official name until 1989 of the Republic of the Union of Myanmar, and *Malachius*, the nominative genus of the tribe.

*Diagnosis*. Body brown-black with a bronze-green metallic lustre; dorsal setae light brown; abdomen black and lacking metallic lustre; conjunction membranes of ventrites yellow. Head prognathous, narrow and strongly protruding distally; eyes round, small, weakly convex; pronotum convex, slightly longer than wide, rectangular, with rounded angles, together with head appearing to be narrowed and elongate. Elytra simple, subparallel, flattened, elongate, evenly expanded behind the middle and rounded posteriorly, completely covering the abdomen. Lateral sides of elytra moderately flattened and edged with carinate rib; epipleurae wide (Figure 2A). Mesepimeres brown; evaginating vesicles yellow-brown. Labial palpi yellow-brown with elongate sub-oval pointed apical palpomere; antennae yellow with three black apical antennomeres, appearing to be ‘clavate’ due to three apical segments widened and elongate; 1st and 2nd antennomeres slightly swollen, subtriangular and of equal size; intermediate segments cylindrical. Legs brown with anterior tarsi, intermediate tibiae and tarsi, and almost all posterior legs yellow. Legs simple, thin, slightly elongate (Figure 1B and Figure 2A); femora of all legs flattened, anterior distinctly widened, and intermediate and posterior thin, not curved; tibiae straight; anterior tibiae widened and slightly curved inwards and sinuate near the apices (Figure 2C); tarsi five-segmented; anterior tarsi depressed; intermediate and posterior tarsi compressed; 4th tarsomere of all legs shorter than other tarsomeres; anterior tarsi wider and shorter; 1st to 3rd tarsomeres simple, as wide as apex of tibiae and almost equal in size to each other; the claw segments are the longest in all legs; claws long, thin, slightly curved and sharp, lacking appendages at base. Wings normally developed. Pygidium (apical tergite) domed, slightly curved ventrally inward; ultimate ventrite (apical sternite) simple, narrow, undivided. Body length 2.18 mm.

*Comparison*. In outward appearance the new genus is similar to the genus *Tanaops* LeConte, 1859, due to its elongate head and narrowed pronotum. The main characteristics which differentiate the new genus are as follows: the 4th tarsomere in all legs is the shortest, while in *Tanaops* the anterior and posterior tarsi have the 4th segment equal in length with the 3rd; the anterior tibiae in *Burmalachius* are widened, slightly curved inwards and excavate near the apices, whereas in *Tanaops* they are simple; in the new genus the tarsomeres of the anterior legs are depressed, with 1st to 3rd tarsomeres simple and almost completely equal in size, whereas in *Tanaops* the anterior tarsi are compressed with the second tarsomere enlarged and slightly prolonged over the 3rd tarsomere, but lacking tarsal comb [29]; the antennae of the new genus appear to be “clavate” due to the three dilated apical antennomeres, with 1st and 2nd antennomeres enlarged and almost equal in size to each other, whereas in *Tanaops* the 2nd antennomere is half the length of the 1st antennomere, the distal antennomeres are not enlarged, and the antennae appear to be serrate due to the triangular shape of the antennomeres.

*Composition*. Only type species.

*Notes*. The new genus definitely belongs to the family Malachiidae of the superfamily Cleroidea due to possessing the typical characteristics listed by Majer [30] as follows: thorax with small but distinct evaginating vesicles; body weakly sclerotized with sternites able to contract and appearing to be flexible; elytra evenly expanded posteriorly and flattened; and males possessing special male characteristics presenting sexual dimorphism (in this case, enlarged and modified tibiae and widened and depressed tarsi in the anterior legs, ‘clavate’ antennae). Provisionally, this genus is placed in the tribe Malachiini based on its 4th tarsomere being distinctly smaller in all legs, but the 2nd antennomere in Malachiidae is half the length of the 3rd antennomere [22], while in *Burmalachius* gen. nov. these two antennomeres are almost similar in size. 

*Burmalachius acroantennatus* sp. nov.

LSID urn:lsid:zoobank.org:act:9E6A6FF0-3008-4C04-91B6-E2A911982CC8 

*Etymology*. The name of the species is composed of two words, the ancient Greek άκρος (*acros*) meaning extreme or distal, and antenna to draw attention to the typical shape of the antennae which is dilated towards the apex. 

*Material*. Holotype ISEZh, no. sch_011, male (sex is confirmed by the apically dilated antennae, the characteristic pygidium which is wide, elongate and dome-shaped, and the dimorphic anterior legs which are dilated, curved and excavate); Burmese amber (or Burmite or Kachin amber originated from the Hukawng Valley in northern Myanmar) dated from the latest Albian to the earliest Cenomanian ages of the mid-Cretaceous period. A small trapezoid piece of amber 8 × 5 × 3 mm in size, with a beetle in dorsal position in the left portion of the piece; remains of an oak flower are also included and located in the right area above the beetle’s head. The small piece of amber with inclusion is disposed in a round plastic transparent box filled with cellular polystyrene.

*Description*. Body brown-black with bronze-green metallic lustre; dorsal setae light brown. Abdomen black lacking metallic lustre; membranous adjoining parts of ventrites yellow. Labial palpi yellow-brown; antennae yellow with three black apical antennomeres. Tarsi in anterior legs, and tibiae and tarsi in intermediate legs yellow; posterior legs almost completely yellow. Mesepimeres brown; evaginating vesicles yellow-brown. Body elongate and sub-oval, widened posteriorly and flattened (Figure 1). Head prognathous, flat, narrow and strongly protruding distally, narrower than pronotum at the base; eyes round, small, weakly convex, without interfacetal setae; frons and interocular area flat and lacking impressions. Surface of head weakly visible, finely punctured, shining, covered with short hairs (Figure 2A,E). Antennae 11-segmented, attached to the head near the anterior edge of the eyes below clypeus, appearing to be “clavate” due to the three dilated apical antennomeres, short, reaching but not extending over the middle of the pronotum. 1st antennomere slightly swollen, subtriangular; 2nd antennomere almost completely the same length and width as the 1st, subtriangular3rd antennomere slightly narrower and shorter that the 2nd; 4th antennomere narrow and small, twice as long as the 2nd, subcylindrical; 5th antennomere narrow, elongate, subparallel, twice as long as the 3rd; 6th and 8th antennomeres 1.5 times as short as the 5th, subcylindrical; 7th antennomere moderately wider and longer than the 6th or 8th, subtriangular; 9th to 11th antennomeres wide and elongate, subtriangular, and club-shaped; 9th antennomere twice as wide and long as the 8th, 1.5 times as wide as the 1st or 2nd, and 1.2 times as long as the 1st and 2nd antennomere; 10th antennomere slightly shorter than the 9th; 11th antennomere equal in length to the 9th, evenly narrowed to the apex; surface of antennae sparsely covered with short erect fine light-brown hairs. Clypeus subquadrate; labrum elongate, protruding, rounded apically. Maxillary palpi elongate, narrow; apical palpomere narrowed and pointed at apex, sub-oval. Left mandible invisible. Right mandible bidentate, rounded, not elongate or protruding. Pronotum convex, slightly longer than wide, rectangular, with rounded angles. Surface evenly and finely punctured, covered with short semi-erect fine hairs. Scutellum poorly visible, transverse, sub-triangular, rounded at apex, with the same puncturing and pubescence as on pronotum. Elytra simple, elongate, evenly expanded behind the middle, slightly convex posteriorly and subparallel, evenly rounded posteriorly, distinctly wider than pronotum at base (Figure 1A and Figure 2A); they completely cover the abdomen. Lateral sides of elytra moderately flattened and edged with carinate rib; epipleurae distinct and wide, finely punctured. Humeri small, indistinct. Elytra with thin suture for whole length of elytra; surface densely and evenly punctured; micro-sculpture indistinct, shining; evenly covered with short semi-erect fine hairs. Legs simple, thin, slightly elongate (Figure 1B and Figure 2B), finely sparsely punctured and covered with short erect hairs visible on outer margin of tibiae; tarsomeres of anterior legs depressed, and intermediate and posterior legs compressed. Femora of all legs flattened; anterior distinctly widened, and intermediate and posterior thin, not curved; tibiae straight; anterior tibiae widened and slightly curved inwards and sinuate near the apices (Figure 2E); intermediate and posterior tibiae thin and straight (Figure 2C,D); all tarsi five-segmented with elongate thin tarsomeres, and 4th tarsomere the shortest in all legs; anterior tarsi wider and shorter; 1st to 3rd tarsomeres simple, as wide as apex of tibiae and almost identical in size; 4th tarsomere twice shorter and narrower; claw segment the longest, somewhat longer that the 1st tarsomere; in intermediate legs, 1st and 2nd tarsomeres elongate, equal in size, subtriangular; 3rd tarsomere twice shorter than the 1st, and of equal size to the 4th; apical segment narrow, distinctly longer that the 1st tarsomere; in posterior legs, 1st to 3rd tarsomeres elongate, subcylindrical, of equal sizes; 4th tarsomere 1.5 times as short as the 1st tarsomere; claw tarsomere distinctly longer that the 1st tarsomere; claws long, thin, slightly curved sharp, lacking appendages at base. 

Wings normally developed. Underside of the body weakly shining, finely and densely punctured and evenly covered with short adpressed fine hairs; mesepimeron elongate, shining; pygidium (apical tergite) domed, slightly curved ventrally inward (due to influence of amber resin); ultimate ventrite (apical sternite) simple, narrow, undivided.

Measurements: length 2.18 mm, width (at the base of the elytra) 0.5 mm.

*Differential diagnosis*. See Comparison under genus description.

## 4. Discussion

Soft-winged flower beetles of the family Malachiidae are distributed worldwide [31,32] and are known to occur in amber inclusions; these beetles are small to moderately sized (1 to 8 mm) with unfused articulated sclerites of the body and characteristic yellow, orange or red evaginate vesicles on the lateral sides of the thorax. The representatives of the family are predators of small invertebrates, and their larvae are found under tree bark [33]. Malachids are seldom found in paleontological materials, but almost every find of fossil beetles represents a new taxon, often without analogues in the recent fauna. Furthermore, paleontological data on Malachiidae are rare, but a short survey is provided by Kirejtshuk and Nel [13], and complemented by Tshernyshev [15,17,22]. The oldest records of soft-winged flower beetles, dating from the late Eocene, refer to several species described from the Florissant lagerstätten formation in Colorado [4], namely, *Collops desuetus* Wickham, 1914, *C*. *extrusus* Wickham, 1914, *C*. *priscus* Wickham, 1914, and *Malachius immurus* Wickham, 1917 (= *Malachius pristinus* Wickham, 1916) [3,4,5,6]. The generic attribution of some of the species mentioned above to the genus *Collops* Erichson, 1840 and the tribe Apalochrini is incorrect, because the 2nd antennomeres in the imprint are round and well developed. The placement of another two species described from this deposit within the Malahiini genera is also problematic, because no *Malachius immurus* Wickham, 1917 (=*Malachius pristinus* Wickham, 1916) or *Cerallus* sp. have characteristics typical for this tribe.

Several species provisionally attributed to *Colotes* have been described from amber inclusions, namely, *Colotes sambicus* Kubisz, 2001 from Baltic amber, and two species, *Colotes constantini* Kirejtshuk et Nel, 2008 and *Colotes impexus* Kirejtshuk et Nel, 2008, from more ancient (earlier Eocene) French amber [13]. Since both species were described from females, generic attribution may be uncertain, and they probably belong to the genus/genera of Palpattalini.

The first Eocenic representative of the tribe Troglopini, *Protocephaloncus perkovskyi*, was described by Tshernyshev [15], who also studied the males of two species which appeared to be similar to *Colotes*. These two species possess characteristics of two tribes, namely, a modified pygidium as in Attalini, and enlarged palpomeres as in Colotini; both were described in a new genus, *Palpattalus* Tshernyshev, 2016, as *P*. *eocenicus* Tshernyshev, 2016 and *P*. *baltiensis* Tshernyshev, 2016. Further study of the Baltic amber revealed a new genus and species, *Palpattalusinus transitivus* Tshernyshev, 2020, which also combined the characteristics of the tribes mentioned above, but it was significantly different from *Palpattalus* Tshernyshev. A new tribe, Palpattalini, for two genera, *Palpattalus* Tshernyshev and *Palpattalusinus* Tshernyshev, and a new representative of the tribe Malachiini, *Premalachius obscurus* Tshernyshev, 2020, were described [17]. Later, a new genus and species, *Aliattalus intercalaris* Tshernyshev, 2021, was described from Baltic amber. The special male characteristics of this species are found in the anterior legs: “… the tibiae of … front legs are laterally compressed and flattened, with a small notch and finely serrated margin along the inner side… The anterior tarsomeres are simp’nj e gbk.kmobrf&le, rather wide, flattened in the dorsoventral direction, second tarsomere without a comb at the top.” [22: 94]. These characteristics are very similar to the special male characteristics found in *Burmalachius acroantennatus* sp. nov.

*Burmalachius acroantennatus* sp. nov. is the first record of the family Malachiidae in Burmese amber and can be considered as the oldest registration of the family in fossil remains. 

To reflect the systematic position of this new species, keys for the identification of Melyrid lineage families, subfamilies of Malachiidae, tribes of the subfamily Malachiinae and genera of the tribe Malachiini are provided below.


**Key to Melyrid lineage families (according to Majer [30]**
**with new modifications)**


1. Ventral side of 2nd and 3rd tarsomeres in anterior tarsi of male densely covered with strong chaetae. Eyes and pronotum distinctly marginate; elytra sparsely covered with semi-erect hairs only. Size 5–18 mm............................................................Prionoceridae

—Ventral side of 2nd and 3rd tarsomeres in anterior tarsi of male simple, lacking strong chaetae. Eyes and pronotum weakly marginate or lacking margination; elytra evenly pubescent, often with double hairs, erect or semi-erect, and adpressed.........................................................….2

2. Thorax with evaginating vesicles; body integument weakly sclerotized; sternites able to contract; adult beetles move rapidly; elytra evenly widened towards the apex and flattened; surface covered with double pubescence; most males possess special male characteristics indicating sexual dimorphism. Size 1–10 mm…............Malachiidae

—Thorax lacking evaginating vesicles; body integument strongly sclerotized; sternites unable to contract; adult beetles cannot move rapidly; elytra evenly rounded towards the apex and weakly flattened; surface covered with erect or adpressed hairs, rarely with double pubescence; most males lack special male characteristics indicating sexual dimorphism …………………………………………………………………………………..…………3

3. 1st tarsomere half the length of the 2nd tarsomere; claws long, with denticle at base; elytra often with longitudinal carinae and regular rows of punctures. Size 5–12 mm ……….....................................Melyridae

—1st tarsomere not shorter than the 2nd tarsomere; claws short, at base with or without appendages; elytra lacking longitudinal carinae, with irregular punctures. Size 3–10 mm................................Dasytidae

In *Burmalachius acroantennatus* sp. nov., the basal tarsomeres of the anterior tarsi are depressed and dilated, but simple, lacking chaetae underside; the thorax has small but visible evaginating vesicles, and the body integument is weakly sclerotized. This species belongs to the subfamily Malachiinae as indicated by the following characteristics: 1st tarsomere of anterior tarsi not widened, and the same shape as the 2nd tarsomere; elytra widened posteriorly and appearing to be sub-oval as a typical “malachid”; all tarsi five-segmented. 


**Key to Subfamilies of Malachiidae**


1. 1st tarsomere of anterior tarsi in male dilated and elongate, typically with a black comb above or hair brash beneath; body elongate and often with uncovered apical segments of abdomen appear to be staphylinoid…….………... Carphurinae Champion, 1923

—1st tarsomere of anterior tarsi in male not dilated and elongate or lacking traces of comb above or hair brash beneath; body elongate or shortened, sub-oval; usually covered apical segment of abdomen; if apical segments of abdomen remain uncovered, it appears to be malachiod, but not staphylinoid ………………………………... 2

2. Elytra parallel, weakly convex and covered with semi-erect elongate hairs, looking like elytra of Dasytidae; evaginating glands very small, but distinct. Anterior tarsi in male compact, with 3rd tarsomere triangular and stretched beneath and 5th tarsomere elongate; possessing brush of chaetae near claws; inner claw twice as long as outer claw …… …..……………………………………………………….…Pagurodactylinae Constantin, 2001

—Elytra parallel or oval, typically covered with short fine pubescence; evaginating glands typical, wide; anterior tarsi in male with simple short claws of equal size; 3rd tarsomere not stretched………………………………3

3. Tarsi appear to be five-segmented; if anterior tarsi in male four-segmented, posterior and intermediate tarsi distinctly five-segmented….…… Malachiinae Fleming, 1821

—Tarsi appear to be four-segmented due to extremely small 1st, 2nd or 4th tarsomere………………………………………………………………………………………………..4

4. Tarsi compact, with extremely short 1st tarsomere; 2nd and 3rd tarsomeres triangular and stretched beneath; 4th tarsomere bilobed, looking like penultimate tarsomere in Cleridae; antennae 11-segmented, slightly dentate and covered with long erect light hairs; elytra shortened, not covering three apical tergites, suboval, weakly convex, habitually looking typically like elytra of Malachiidae………………….… Attalomiminae Majer, 1994

—Tarsi with simple, distinct 1st tarsomere and extremely small 2nd or 4th tarsomere...........................................................5

5. Elytra simple, not appendiculate, sub-oval, weakly convex, evenly rounded apically. Antennae simple, 11-segmented; intermediate antennomeres slightly dentate; apical oval and pointed, bare or weakly pubescent; tarsi five-segmented, compact, with extremely short 4th tarsomere; 1st tarsomere in posterior legs triangular, stretched beneath with tuft of hairs apically; claws simple, short......................…… Amalthocinae Majer, 2002

—Elytra simple, not appendiculate, parallel or sub-oval, weakly convex, evenly rounded apically. Antennae simple, 11-segmented; intermediate antennomeres slightly triangular with stretched outer edges; apical oval or slightly pointed, weakly pubescent; tarsi with extremely small 2nd tarsomere, so that they appear to be four-segmented, 3rd and 4th tarsomeres depressed, expanded and flattened beneath, and appearing to be bilobed; claws simple, short.................................................................. Lemphinae Wittmer, 1976

The next step is to identify which tribe the new species belongs to. Recently, a new method of identifying tribes based on the shape of tarsomeres and antennomeres was proposed by Tshernyshev [34] as follows:


**Key to the tribes of soft-winged flower beetles of the subfamily Malachiinae**


1. 2nd antennomere small, invisible, almost completely hidden by the 1st antennomere.................. Apalochrini

—2nd antennomere distinctly visible …………………..………………………………. 2

2. 2nd antennomere half or distinctly shorter than the 3rd antennomere.…………… 3

—2nd antennomere similar or slightly narrower than the length of the 3rd antennomere........................................4

3. 4th tarsomere of all legs the shortest; 2nd antennomere half the length of the 3rd antennomere; if not, then 1st and 2nd antennomeres modified and enlarged......................................................................Malachiini

—4th and 3rd tarsomeres equal in length in anterior or middle legs; 2nd antennomere very small and less than half the length of the 3rd antennomere ...........................Illopini

4. 2nd antennomere equal to or slightly shorter than the 3rd antennomere in length; 4th and 3rd tarsomeres of anterior legs of similar length...…………………….…... Ebaeini

—2nd antennomere distinctly shorter than the 3rd antennomere.….………………5

5. 4th tarsomere of anterior legs shorter than the 3rd tarsomere …………........Attalini

—4th tarsomere of anterior legs absent or of similar length to the 3rd tarsomere…. 6

6. 1st tarsomere of middle and posterior legs shorter or same length as the 2nd tarsomere………………….…… Troglopini

—1st tarsomere of middle and posterior legs wider and longer than the 2nd tarsomere………………......... Colotini

According to the above Key, *Burmalachius acroantennatus* sp. nov. belongs to the tribe Malachiini since the 4th tarsomere is the shortest in all legs. The ancestral form, Burmalachius, demonstrates a transition of the enlarged first two segments for further modification to special male characteristics as it is typical of the genera *Malachius* F., *Micrinus* and others. Thus, the newly registered species and genus, *Burmalachius* acroantennatus, from Burmese amber should be attributed to the Malachiini tribe of the subfamily Malachiinae. Amongst the genera in the tribe the new genus can be placed near *Acromalachius* Evers.


**Key to genera of the tribe Malachiini**


1. Male special structures indistinct....................................................................................2 

—Male special structures distinct, located on antennae, head, legs or elytral apex...6 

2. Claws with small indistinct membrane at base ………….…………….……………. 3

—Claws with distinct membrane at base; apical segment of palp slender, cylindrical …………………….. 4

3. Apical segment of palp cylindrical; antennae distally not serrated; 2nd antennomere shorter than the 3rd; elytra ovoid, with longitudinally elongated humeri; pronotum not narrowed to the base and convex posteriorly......….*Chionotopus* Abeille de Perrin, 1881

—Apical segment of palp fusiform; antennae distally serrated; 2nd antennomere the same length as the 3rd; elytra broadly widened posteriorly not ovoid, with tuberculose-convex humeri; pronotum narrowed to the base and depressed posteriorly……………………………………………………………...*Premalachius* Tshernyshev, 2020

4. Pronotum longitudinal; antennae always filiform.............. *Microlipus* LeConte, 1852

—Pronotum transverse; antennae filiform, slightly serrate or pectinate……..………..5

5. Antennae filiform or slightly serrate.…*Haplomalachius* (*Haplomalachius*) Evers, 1985 

—Antennae pectinate……………...…. *Haplomalachius* (*Flabellomalachius*) Evers, 1985

6. Antennae dilated to the apex………………………………..…………………………. 7

—Antennae not dilated to the apex…………………………………………………...…. 8

7. Anterior legs of male with simple tibiae and compressed tarsi; 2nd antennomere shorter and narrower than the 1st................................................… *Acromalachius* Evers, 1985

—Anterior tibiae of male widened, slightly curved inwards and excavate near the tip; basal tarsomeres in anterior legs depressed and widened; 1st and 2nd antennomeres enlarged and of identical size……………………………………….......... *Burmalachius* gen. nov.

8. Male special structures located in elytral apex as impressions or appendages….....9

—Male special structures located on head as excavations or protuberances and/or in basal antennomeres ……………………………………………………………………….........20

9. 2nd segment of anterior tarsi in male with a comb overhang; elytral apices in male strongly impressed, with lamellate vertical appendages…*Axinotarsus* Motschulsky, 1854

—2nd segment of anterior tarsi in male simple, lacking comb; elytral apices in male strongly impressed, with spear-shaped or spicula-shaped appendages …………..………10

10. Pronotum longitudinal, extremely narrowed posteriorly...............................…….11

—Pronotum more or less equilateral, not sinuate posteriorly..............………………. 13

11. Beetles 4–4.5 mm in length; elytral impression in male deep; appendage black, strong, spear-shaped…………………………………………………………..……….……… 12

—Beetles 3–3.5 mm in length; elytral impression in male not deep; appendage yellowish or black, fine, spicula-shaped……………………….……….. *Charopus* Erichson 1840

12. 3rd antennomere inversely triangular with stretched lower edge; 6th antennomere simple…………………………………...……….. *Cyrtosus* (*Cyrtosus*) Motschulsky 1854

—3rd antennomere simple; 6th antennomere elongate and with stretched outer edges with a bi-horned appearance...................…….……. *Cyrtosus* (*Oogynes*) Mulsant & Rey 1867

13. Elytral apices in male impressed dorsally; distal margin stretched and sinuate, bearing bunches of hairs…..........................................………. *Cerapheles* Mulsant & Rey 1867

—Elytral apices in male impressed frontally; distal margin simple, not stretched and sinuate, lacking bunches of hairs.................................................................................………. 14

14. 1st antennomere swollen, cylindrical; apical tergite bisinuate on distal part, with bunches of hairs.................................................................... *Clanomalachius* Tshernyshev, 2000

—1st antennomere not swollen; apical tergite with evenly rounded distal part, bunches of hairs lacking ……………………………………………..…………..…………… 15

15. 1st antennomere parallel; intermediate antennomeres with slightly stretched apical sides; elytral impression with horn-like appendage or lacking it...................................16

—1st antennal segment clavate; intermediate antennomeres triangular; elytral impression with spear-shaped or lamellate appendage………………………………………. 17

16. Elytral apices strongly impressed, with black strong appendage inside…………………...*Clanoptilus* (*Clanoptilus*) Motschulsky, 1853

—Elytral apices slightly depressed, with or without small appendage……………………………...………...……*Clanoptilus* (*Hypoptilus*) Mulsant and Rey 1867

17. Intermediate antennomeres with slender apical sides, parallel; elytral impression with or without a lamellate appendage………….……. *Anthomalachius* Tshernyshev, 2009

—Intermediate antennomeres 3–10 triangular; elytral impression in male with curved lamellae and/or spear-shaped black appendage…………………………………… 18

18. Elytral impression in male with dorsally curved lamellae at apex and spear-shaped black appendage inside….. ……….*Anthocomus* (*Omphalius*) Abeille de Perrin, 1891

—Elytral impression in male with spear-shaped black appendage and laterally curved lamellae……………………………………………………….…………………………19

19. Hind tibia in male simple, without indentation……………………………………………...……*Anthocomus* (*Anthocomus*) Erichson, 1840

—Hind tibia in male with indentation before the apex…………………………………………...……*Anthocomus* (*Celidus*) Mulsant & Rey, 1867

20. Male head excavation lacking…................................................................................. 21

—Male head with transverse excavation that is deeply or slightly impressed......... 23

21. 1st and 2nd antennomeres enlarged and modified.. *Micrinus* Mulsant & Rey, 1867

—1st and 2nd antennomeres not enlarged and modified, and the 1st antennomere distinctly larger than the 2n……………………………………………………….…………. 22

22. Male head with smooth protuberance between antennae; 5th antennomere simple, not enlarged or modified.............................................................*Cordylepherus* Evers, 1985

—Male head with strong horn between antennae which is round and cut at the tip; 5th antennomere large, funnel-shaped............................ *Ceratistes* Fisher de Walheim, 1844 

23. Male head excavation located behind the antenna in a distal part of head; protuberance inside the excavation bearing long hairs; antenna with appendages.................... 24

—Male head excavation located before the antenna; distal part lacking protuberance; antenna simple; appendages lacking……….….... *Anhomodactylus* Mayor & Wittmer, 1981 

24. Male head excavation deep……………...… *Malachius* (*Malachius*) Fabricius, 1775

—Male head excavation very slightly depressed………………………………………………….*Malachius* (*Protomalachius*) Evers, 1985

## 5. Conclusions

Cleroidea fossil representatives of the Melyridae group are well known from Eocene deposits, i.e., in French, Baltic and Rovno ambers and in imprints of the Florissant deposit [1,2,3,4,5,6,7,8,9,10,11,12,13,14,15,16,17,18,19]. These beetle records date from 66 to 23 million years ago, in the Eocene Epoch, and were the oldest for the Malachiidae family. The Melyridae lineage of the Cleroidea is mentioned in publications devoted to Burmese amber [35,36,37,38] and imprints found in Yixian Formation (Barremian, Lower Cretaceous, some 125 million years ago) from Liaoning [39], where the fossils are described as follows: “One print found looked like a member of Melyridae Leach, 1815. Another Cleroid print has the dentate tibiae somewhat reminiscent of tibiae of recent members of Dasytidae Laporte de Castelnau, 1840.” Family attribution of the species mentioned from these deposits as “Melyridae sensu lato” remains undefined. The first record of the *Burmalachius acroantennatus* sp. nov. in Burmese amber provided evidence to show that the family Malachiidae was formed much earlier than the Eocene Epoch, as was previously known, at about 100 million years ago, from the latest Albian to earliest Cenomanian ages of the mid-Cretaceous period [20,40,41,42,43,44,45,46,47]. At this time characteristics typical for Malachiidae were formed, namely, the small evaginating vesicles in the thorax, the apically dilated antennae with two widened basal antennomeres, the modified tibiae and tarsi of the anterior legs, and the domed curved pygidium, which are characteristic to *Burmalachius* and can be found in a number of recent species. 

The fact that Dasytidae are found in Charentese amber [19], Prionoceridae in Burmese amber [21] and Malachiidae in Burmese amber is clear evidence of the ancient history of these families, which has sustained their independent existence for 100 million years from the Cretaceous Epoch to the present day. Such a long existence of independent groups provides a new argument for their systematic positioning as separate families.

## Data Availability

The specimens are deposited in S.E. Tshernyshev’s collection at the Institute of Systematics and Ecology of Animals, Siberian Branch, Russian Academy of Sciences, Novosibirsk.

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
