# Peer review of "First Record of the Family Malachiidae (Coleoptera: Cleroidea) from Mid-Cretaceous Burmese Amber with a Description of Burmalachius acroantennatus Gen. et Spec. Nov."

_life, 2023, doi:10.3390/life13091938_

Round 1

Reviewer 1 Report

This MS is an ordinary report to the finding of a new amber taxon from Myanmar, a Malachiid new genus et  new species. The paper gives a  clear result of authors' study and describes the new taxon in a relatively high quality. Enough messages are given in the MS about methods and type material; ZooBank registrations for the name of the new taxon (genus et species) is well done and lists out in the MS. 

The conclusion for this MS is accept in present form. 

Author Response

Dear colleague,

we are very much grateful for your time you wasted to review our manuscript! Thank you for the kind words regarding our work. 

Thank you again!

With the kindest wishes,

Sincerely,the authors

Reviewer 2 Report

Authors present a new finding of the Malachiidae beetle from the Myanmar amber, and in general the paper is interesting, however, it requires careful english corrections preferably by the native speaker as grammatical structure of some parts of the text is clearly not properly checked. Other detailed comments are marked in the attached pdf file. In several aspects manuscript needs improvements e.g. in teh description at generic and specific levels (both description should not double the same sentences, generic and specific characters should be split), color illustrations can be improved, enlarged photos of the habitus, even the quality is not excellent, would be much more informative than currently presented, because they are simply too small to see anything. In my opinion the key to all Melyridae attached to this paper is unnecessary, this is not a review of the whole family. Another thing concerns the origin of the specimen which is here examined. I am not supporting new tendencies that authors should present a government certificate, however, the source of the fossil is very important, and because authors do not provide the ft-ir spectrum in my opinion clear indication of the origin of that specimen is mandatory. I recommend minor revision of that MS.

Author Response

Authors present a new finding of the Malachiidae beetle from the Myanmar amber, and in general the paper is interesting, however, it requires careful english corrections preferably by the native speaker as grammatical structure of some parts of the text is clearly not properly checked. Other detailed comments are marked in the attached pdf file. In several aspects manuscript needs improvements e.g. in teh description at generic and specific levels (both description should not double the same sentences, generic and specific characters should be split), color illustrations can be improved, enlarged photos of the habitus, even the quality is not excellent, would be much more informative than currently presented, because they are simply too small to see anything. In my opinion the key to all Melyridae attached to this paper is unnecessary, this is not a review of the whole family. Another thing concerns the origin of the specimen which is here examined. I am not supporting new tendencies that authors should present a government certificate, however, the source of the fossil is very important, and because authors do not provide the ft-ir spectrum in my opinion clear indication of the origin of that specimen is mandatory. I recommend minor revision of that MS.

Dear colleague,

thank you so much for precise reviewing of our manuscript, the improved version prepared according to your recommendation looks much better. We added necessary paragraphs, i.e. the part regarding thermilogy. Some of the paragraph, such as Comparison, Composition and Notes  were prepared according to the traditional paleontological descriptions and a rules of the Life and are remained as was initially proposed.

Regarding necessity of the Keys provided in the manuscript, other reviewers pointer out the following: “… The most interesting section surely is an attempt to provide keys for identification of Melyrid lineage families, subfamilies of Malachiidae, tribes of the subfamily Malachiinae and genera of the tribe Malachiini”. In fact, this is a fist compilation of the keys most of which are presented for the first time. We found them an import part of the work aimed to pay attention for characters diversity in Melyridae lineage beetle families.

Also, we remained some terms commonly used in descriptions of Malachiidae to provide continuity of works. Also, some terms and linguistic phrases were proposed by Prof Mark Seaward, who  revised English text of our manuscript.

We are grateful for your work on our manuscript.

Thank you for your time!

Sincerely,

the authors

For descriptions, special male structures and genitalia were studied. The term “special male structures” is not analogue to the term “Excitatoren”, that means different kind of structures located in different part of male body of soft winged flower beetles and bearing ducts of pheromone glands necessary for female attraction and successful copulation (Evers 1956, 1963, 1988; Matthes 1962). The “special male structures” includes all typical parts of male, irrespective of providing them with pheromone glands or not. In the new species widened and emarginate anterior tibiae and dilates basat tarsomeres in anterior legs, and enlarged apical antennomeres are considered as special male characters. Terminology of terminalia morphology is according to Lawrence et

  1. al. (2010), namely (in comparison with previously used terms): pygidium for apical tergite, ultimate abdominal

ventrite for apical sternite

  1. Evers, A.M.J. Uber die Funktion der Excitatoren beim Liebesspiel der Malachiidae. (11. Beitrag zur Kenntnis der Malachiidae).

Entomol. Bl. Biol. Syst. Kafer 1956, 52, 165–169.

  1. Evers, A.M.J. Uber die Entstehung der Excitatoren und deren Bedeutung fur die Evolution der Malachiidae (Col.). (19. Beitrag

zur Kenntnis der Malachiidae). Acta Zool. Fenn. 1963, 103, 1–24.

  1. Evers, A.M.J. Zur Evolution von Koadaptationen. Die Excitatoren bei den Malachiidae. Entomol. Bl. 1988, 84, 61–66.
  2. Matthes, D. Excitatoren und Paarungsverhalten Mitteleuropaischer Malachiiden (Coleopt., Malacodermata). Z. Morphol. Okologie Tiere 1962, 54, 375–546. [http://doi.org/10.1007/BF00406263]

Lawrence, J.F., Beutel, R.G., Leschen, R.A.B. & Ślipiński, S.A. (2010) Chapter 2. Glossary of Morphological Terms. In:

Leschen, R.A.B., Beutel, R.G. & Lawrence, J.F. (Eds.), Handbook of Zoology, Coleoptera. Vol. 2. Morphology and

Systematics (Elateroidea, Bostrichformia, Cucujiformia partim). Walter de Gruyter, Berlin, pp. 9–20.

https://doi.org/10.1515/9783110911213.9

Reviewer 3 Report

see attached files

and here 

In this paper the authors describe a new genus of Malachiidae based on a new species found in Burmese amber. As reported by the authors this is the first record of the family Malachiidae in Burmese amber. The description of both genus and species seems somewhat accurate although unfortunately based on a single specimen.

The most interesting section surely is an attempt to provide keys for identification of Melyrid lineage families, subfamilies of Malachiidae, tribes of the subfamily Malachiinae and genera of the tribe Malachiini (this must be added in the Abstract). Unfortunately in many dichotomies the characters in opposition are often missing or inadequate (I have highlighted several of them). Therefore the keys must be revised accordingly.

In the descriptions and in the keys sometimes the authors correctly use the present participle of the verb but just below they use the verb to the present. This must be revised. The same is for the articles.

I think that several little changes are necessary, e.g. (other modifications are suggested in the text):

line 14. to the tribe Malachiini (Coleoptera: Malachiidae), is described from

line 45. Malachiid imagoes differ in their morphology, the males of most species being pro-

line 50. Tarsomeres of the anterior tarsi being wide and depressed.

line 217. [26,27] and are known to occur in amber inclusions; these beetles are small to moderately

line 224. Kirejtshuk and Nel [13], and complemented by Tshernyshev [15,17,22]. The

line 243. appeared to be similar to Colotes. These two species possess characters of two tribes,

line 249. ferent from Palpattalus [please remind that the name of the author has to be reported only when the taxon is cited for the first time]. A new tribe….

Line 466. Family attribution of the species mentioned from these deposits as “Melyridae sensu lato

Author Response

see attached files

and here

In this paper the authors describe a new genus of Malachiidae based on a new species found in Burmese amber. As reported by the authors this is the first record of the family Malachiidae in Burmese amber. The description of both genus and species seems somewhat accurate although unfortunately based on a single specimen.

The most interesting section surely is an attempt to provide keys for identification of Melyrid lineage families, subfamilies of Malachiidae, tribes of the subfamily Malachiinae and genera of the tribe Malachiini (this must be added in the Abstract). Unfortunately in many dichotomies the characters in opposition are often missing or inadequate (I have highlighted several of them). Therefore the keys must be revised accordingly.

We have added all necessary changes in the Keys, but several point remained due to recommendation of our English editor. Thus, our expression “… looks like…” Prof. Seaward recommended substitute by “ appear to be” as more correct in English grammar.

In the descriptions and in the keys sometimes the authors correctly use the present participle of the verb but just below they use the verb to the present. This must be revised. The same is for the articles.

I think that several little changes are necessary, e.g. (other modifications are suggested in the text):

line 14. to the tribe Malachiini (Coleoptera: Malachiidae), is described from

done

line 45. Malachiid imagoes differ in their morphology, the males of most species being pro-

done

line 50. Tarsomeres of the anterior tarsi being wide and depressed.

done

line 217. [26,27] and are known to occur in amber inclusions; these beetles are small to moderately

done

line 224. Kirejtshuk and Nel [13], and complemented by Tshernyshev [15,17,22]. The

done

line 243. appeared to be similar to Colotes. These two species possess characters of two tribes,

done

line 249. ferent from Palpattalus [please remind that the name of the author has to be reported only when the taxon is cited for the first time]. A new tribe….

sorry, this depends of the Journal rules, we prepared the text according to the Life Journal requirements

Line 466. Family attribution of the species mentioned from these deposits as “Melyridae sensu lato”

done

Thank you so much for precise reviewing of our manuscript, the improved version prepared according to your recommendation looks much better. Thank you for your time!

Sincerely,

the authors

Round 2

Reviewer 3 Report

OK. the paper can be published in the present version.